**Data Availability Statement:** The minimal anonymized Swiss and Polish dataset are available on Zenodo (https://zenodo.org/, DOI: 10.5281/zenodo.5024262). Data were collected from

# Risk factors for treatment failure in women with uncomplicated lower urinary tract infection

Romain Martischang[1]*, Maciek Godycki-Ćwirko[2], Anna Kowalczyk[2], Katarzyna Kosiek[3], Adi Turjeman[4,5], Tanya Babich[4,5], Shachaf Shiber[6], Leonard Leibovici[4,5], Elodie von Dach[7,8], Stephan Harbarth[1,7], Angela Huttner[1,7]

1 Infection Control Program and WHO Collaborating Centre on Patient Safety, University of Geneva Hospitals and Faculty of Medicine, Geneva, Switzerland, 2 Centre for Family and Community Medicine, Faculty of Health Sciences, The Medical University of Lodz, Lodz, Poland, 3 Family Doctors' Clinic Pomorska96, Lodz, Poland, 4 Sackler Faculty of Medicine, Tel Aviv University, Tel Aviv, Israel, 5 Department of Internal Medicine E, Rabin Medical Center, Beilinson Campus, Petah-Tiqva, Israel, 6 Department of Emergency Medicine, Rabin Medical Center, Beilinson Campus, Petah-Tiqva, Israel, 7 Division of Infectious Diseases, University of Geneva Hospitals and Faculty of Medicine, Geneva, Switzerland, 8 Center for Clinical Research, University Hospitals of Geneva, Geneva, Switzerland

* romain.martischang@hcuge.ch

## Abstract

Given rising antibiotic resistance and increasing use of delayed prescription for uncomplicated lower urinary tract infections (UTI), patients at risk for treatment failure should be identified early. We assessed risk factors for clinical and microbiological failure in women with lower UTI. This case-control study nested within a randomized clinical trial included all women in the per-protocol population (PPP), those in the PPP with microbiologically confirmed UTI, and those in the PPP with UTI due to *Escherichia coli*. Cases were women who experienced clinical and/or microbiologic failure; controls were those who did not. Risk factors for failure were assessed using multivariate logistic regression. In the PPP, there were 152 clinical cases for 307 controls. Among 340 women with microbiologically confirmed UTI, 126 and 102 cases with clinical and microbiological failure were considered with, respectively, 214 and 220 controls. Age ≥52 years was independently associated with clinical (adjusted OR 3.01; 95%CI 1.84–4.98) and microbiologic failure (aOR 2.55; 95%CI 1.54–4.25); treatment with fosfomycin was associated with clinical failure (aOR 2.35; 95%CI 1.47–3.80). The association with age persisted among all women, and women with *E. coli*-related UTI. Diabetes was not an independent risk factor, nor were other comorbidities. Postmenopausal age emerged as an independent risk factor for both clinical and microbiological treatment failure in women with lower UTI and should be considered to define women at-risk for non-spontaneous remission, and thus for delayed antibiotic therapy; diabetes mellitus was not associated with failure.

Geneva (Switzerland), Lodz (Poland), and Tel Aviv (Israel). By Swiss and Polish laws, there is no restriction to share anonymized information. If individual researchers want to recycle the complete dataset using the Israeli dataset, this additional information can be made available upon reasonable request.

**Funding:** The author(s) received no specific funding for this work.

**Competing interests:** The authors have declared that no competing interests exist.

## Introduction

Acute, uncomplicated lower urinary-tract infection (UTI) is one of the most frequent indications for antibiotic prescription among healthy women [1]. With their widespread use, there is increasing resistance to fosfomycin and nitrofurantoin [2], the two antibiotics currently recommended as first-line therapy [1]. At the same time, delayed antibiotic treatment strategies [3] are being increasingly used for this mucosal infection given its sizeable rate of spontaneous remission [4, 5], the low risk of progression to pyelonephritis [1], and clear evidence that frequent antimicrobial therapy increases the risk of acquiring multi-resistant organisms [6, 7]. As resistance to first-line agents increases and delayed-therapy approaches become more popular, there is a growing need to identify appropriate candidates by identifying risk factors for clinical failure or non-recovery. Fortunately, pyelonephritis remains a rare complication among healthy women [8, 9]. Age has been associated with treatment failure in multiple observational studies, sometimes yielding conflicting results [4, 10, 11]. Other baseline factors have been identified inconsistently, including diabetes, chronic kidney disease, previous hospitalization, history of recurrent cystitis, and increased comorbidity [11–15], as have particular symptoms, such as frequency and urgency [16]. Treatment failure, however, was defined heterogeneously, and follow-up bacteriologic data were rarely available [15].

From 2013 to 2017, we conducted the multicentre randomized clinical trial (RCT) "AIDA", which showed clinical and microbiologic superiority of nitrofurantoin over single-dose fosfomycin in non-pregnant women with acute, uncomplicated lower UTI [17]. Here we report the results of a nested case-control study assessing the influence of age and other potential risk factors for clinical and microbiologic failure in the AIDA trial [18].

## Methods

The AIDA study was an open-label/analyst-blinded, multicentre RCT conducted from 2013 to 2017 in Geneva (GE), Switzerland; Lodz (LO), Poland; and Petah-Tiqva (TA), Israel. It included 513 hospitalized and ambulatory adult non-pregnant women with lower UTI symptoms and a positive urine dipstick test. Those with suspected upper UTI, recent or ongoing antibiotic use, immunosuppression, severe renal insufficiency, indwelling urinary catheter or otherwise complicated UTI were excluded. Participants were randomly assigned to macrocrystalline nitrofurantoin 100 mg 3 times a day for 5 days or a single 3-g dose of oral fosfomycin and followed clinically and microbiologically at 14 (±2) and 28 (±7) days after completion of antibiotic therapy. The per-protocol population (PPP) consisted of women with at least 80% medication adherence, no major protocol deviations, and available primary-outcome data (474/513, 92.4%). This study was performed in accordance with the STROBE statement for case-control studies (S1 Appendix).

This nested case-control study assessed three populations: (1) all women randomized in the AIDA study and adhering to the study protocol (PPP) [17], (2) all women in the PPP with microbiologically confirmed ("culture-positive") UTI, and (3) all women in the PPP with UTI due to *Escherichia coli*. Only women adhering to the RCT's protocol were included to avoid the problems of missing data/outcomes and non-adherence. Assuming that *E. coli* are almost never contaminants in acute cystitis, this third cohort represents the best "confirmation" of a true UTI, excluding all results of lower quality with mixed flora and potential contaminants [19]. Patients with indeterminate clinical outcomes were excluded from all populations, and patients with negative or missing cultures at baseline were excluded from the second and third populations.

In the present report, the first population (PPP) was assessed to evaluate risk factors for clinical failure. The second and third populations were assessed to evaluate risk factors for

both microbiologic and clinical failure. Case patients in the nested study were subclassified into "clinical" and "microbiologic", defined respectively as those who experienced clinical and/ or microbiologic failure (as defined previously) [17] in the 28 days following therapy completion; all available controls were included, defined according to case definitions as women either with clinical or microbiologic success. Women with missing bacteriological outcomes were excluded.

We assessed the impact on failure of demographic, clinical, and microbiologic characteristics defined previously (Appendix I). The association between the infecting pathogen and clinical and microbiological failure was explored in sensitivity analyses, adjusting for both presence of *E. coli* at baseline and resistance to the study drug received. The relationship between persistent, asymptomatic bacteriuria and subsequent clinical failure was also assessed in exploratory analyses.

### Statistical analysis

Continuous variables are reported as the mean (±SD) or median (IQR) according to their distribution. Baseline characteristics were compared using $X^2$, *t*-test or Wilcoxon test. LOESS smoothing function, a locally weighted non-parametric scatterplot technique, was used to explore variation in clinical failure by age and treatment and to categorize continuous variables, as described elsewhere [20]. We defined an age cut-off of 52 years, as the average age of menopause is considered physiologically to be between 50 and 52 years [21]. The effects of this cut-off can be observed graphically in the (S1 Fig in S1 Appendix).

Two multivariate logistic regression models assessed associations between demographic and clinical characteristics and clinical and microbiological failure through day 28 in those with culture-positive UTI. In exploratory analyses, similar models including the same variables assessed clinical failure in the PPP population, and clinical and microbiological failure in the *E. coli* subgroup, through day 28. Candidate variables for the multivariate models were chosen according to the "best model" following Bayesian Model Averaging (BMA) methods [22, 23]. Potential confounders were tested stepwise in the multivariate models, and retained if the change they induced on the beta coefficient was >20%, as defined previously [24]. Variables were examined for collinearity using a collinearity matrix. Analyses were performed using R.3.6, and RStudio Team (2015), the package "BMA" for Bayesian Model Averaging [25], and "MASS" for the creation of generalized linear models [26]. The AIDA study was approved by the Geneva Cantonal Ethics Commission (13–014) and Swissmedic (2013DR4095). Using only anonymized information, this nested study did not require further ethical approval.

### Results

Among the 513 women randomized, 459 remained in the PPP and had a determinate outcome, 340 (66%) had culture-positive UTI, and 204 of these (60%) had *E. coli* infections (Fig 1). In the PPP, there were 152 clinical cases for 307 controls. Among the 152 cases, five had pyelonephritis. Among those with culture-positive UTI, there were 126 and 102 clinical and microbiologic case patients for 214 and 220 controls, respectively. In the *E. coli* subgroup, there were 70 and 71 clinical and microbiologic case patients for 134 and 128 controls. Treatment received was balanced among cases and controls (Table 1).

Women who experienced clinical or microbiologic failure by day 28 were on average 10 years older (mean age 56.8 ±22.5 and 58.0 ±20.8 versus 45.8 ±19.6 and 47.2 ±20.7 years, respectively, [p<0.001]); the same pattern was seen in the *E. coli* subgroup (Fig 2, S1 Fig in S1 Appendix). In crude comparisons, women with clinical failure were significantly more likely to present with urgency and chills, to have an increased risk for carriage of multidrug-resistant

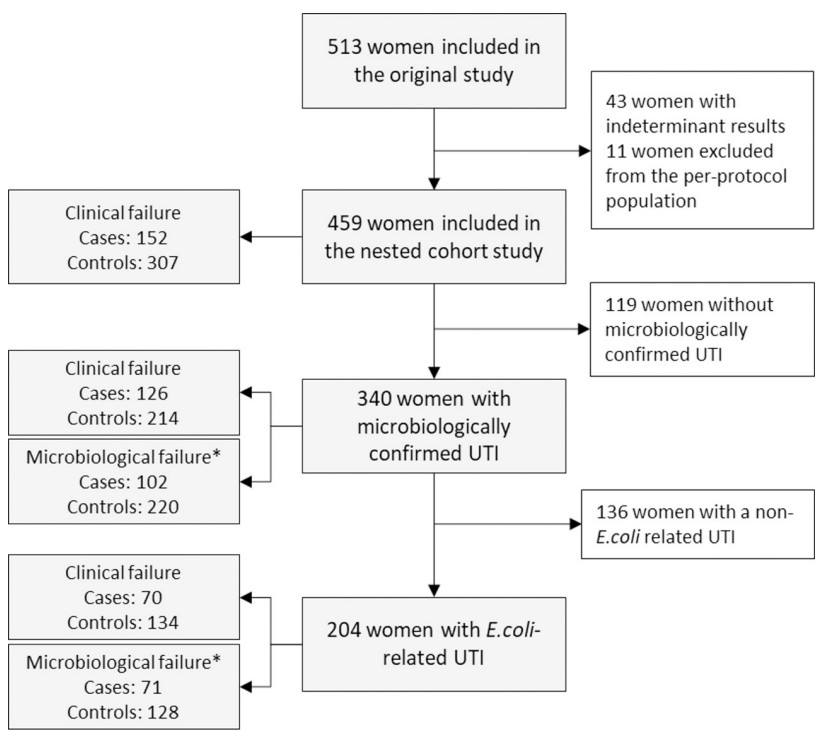

**Fig 1. Study flowchart.** * Negative or missing cultures at baseline were excluded.

bacteria, to be treated with fosfomycin, and to be recruited at the Tel Aviv site (Table 1, S1-S5 Tables in S1 Appendix). We had no missing data. The 54 participants excluded from this analysis almost shared the same characteristics (S6 Table in S1 Appendix). In the per-protocol and *E. coli* populations, associations between clinical failure, age and fosfomycin persisted (S2-S5 Figs in S1 Appendix). Women with microbiologic failure were more likely to have diabetes mellitus and a strongly positive baseline dipstick result (with both nitrites and leucocyte esterase detected), and to be recruited in TA (Table 1).

Following BMA, the most parsimonious model retained age, fosfomycin treatment, and recruitment site (TA) as independent predictors for clinical failure, and age for microbiologic failure (S2-S4 Figs in S1 Appendix). Those aged >52 years had an increased risk for clinical and microbiologic failure with adjusted odds ratios (aOR) of 3.01 (95%CI 1.84–4.98) and 2.55 (95%CI 1.54–4.25), respectively (Table 2). The risk was even higher in the *E. coli* subgroup (aOR 4.03 [95%CI 2.05–8.18] and 3.08 [95%CI 1.64–5.87] for clinical and microbiologic failure respectively, S7 and S8 Tables in S1 Appendix). When applying the same model to the overall PPP population, the effect of age > 52 years on risk of clinical failure persisted (aOR 3.07 [95% CI 2.01–4.75], S9 Table in S1 Appendix). In patients >52 years old, nitrofurantoin remained superior to fosfomycin in terms of clinical, but not microbiologic, success (Table 2, S1 Fig in S1 Appendix). In Bayesian model averaging, diabetes mellitus did not emerge as a risk factor for clinical or microbiological failure. These observations remained when adjusting for the presence of *E. coli* and resistance to the study drug (S10 and S11 Tables in S1 Appendix).

In all women with positive baseline cultures (n = 352), the predominant isolates were *E. coli* (n = 204), mixed flora (n = 47), *Klebsiella pneumoniae* (n = 23), *Proteus spp.* (n = 13), and Gram-positive flora (n = 17). No significant variation was observed for these pathogens across different age categories (Fig 3). Co-pathogens were not considered in this analysis (n = 51). Seventeen (9%) women treated with nitrofurantoin and four (2%) treated with fosfomycin had

**Table 1. Population characteristics.**

| | Clinical Failure | | | Microbiological Failure | | |
|---|---|---|---|---|---|---|
| | Cases (n = 126) | Controls (n = 214) | p-value | Cases (n = 102) | Controls (n = 220) | p-value |
| **Age >52 years (%)** | 75 (59.5) | 74 (34.6) | **<0.001** | 64 (62.7) | 82 (37.3) | **<0.001** |
| **Dysuria (%)** | 94 (74.6) | 174 (81.3) | 0.19 | 74 (72.5) | 181 (82.3) | 0.06 |
| **Frequency (%)** | 116 (92.1) | 185 (86.4) | 0.16 | 95 (93.1) | 192 (87.3) | 0.17 |
| **Urgency (%)** | 106 (84.1) | 158 (73.8) | **0.04** | 85 (83.3) | 167 (75.9) | 0.17 |
| **Supra-pubic discomfort (%)** | 60 (47.6) | 102 (47.7) | 1 | 43 (42.2) | 104 (47.3) | 0.46 |
| **Gross hematuria (%)** | 23 (18.3) | 32 (15.0) | 0.52 | 11 (10.8) | 38 (17.3) | 0.18 |
| **Flank pain (%)** | 16 (12.7) | 17 (7.9) | 0.21 | 12 (11.8) | 18 (8.2) | 0.41 |
| **Chills (%)** | 21 (16.7) | 19 (8.9) | **0.05** | 12 (11.8) | 22 (10.0) | 0.78 |
| **Recurrent UTI* (%)** | 24 (19.0) | 27 (12.6) | 0.15 | 17 (16.7) | 32 (14.5) | 0.74 |
| **Diabetes mellitus (%)** | 18 (14.3) | 16 (7.5) | 0.07 | 18 (17.6) | 16 (7.3) | **0.01** |
| **Fosfomycin (%)** | 78 (61.9) | 87 (40.7) | **<0.001** | 58 (56.9) | 99 (45.0) | 0.06 |
| **Inclusion in Geneva (%)** | 43 (34.1) | 104 (48.6) | **<0.01** | 38 (37.3) | 108 (49.1) | **0.01** |
| **Inclusion in Lodz (%)** | 49 (38.9) | 83 (38.8) | | 55 (53.9) | 77 (35.0) | |
| **Inclusion in Tel-Aviv (%)** | 34 (27.0) | 27 (12.6) | | 9 (8.8) | 35 (15.9) | |
| **Risk score for Resistance = 0 (%)*** | 15 (11.9) | 36 (16.8) | **0.05** | 14 (13.7) | 36 (16.4) | 0.79 |
| **Risk score for resistance = 1 (%)*** | 94 (74.6) | 132 (61.7) | | 69 (67.6) | 141 (64.1) | |
| **Risk score for resistance = 2 (%)*** | 17 (13.5) | 46 (21.5) | | 19 (18.6) | 43 (19.5) | |
| Nitrites on dipsticks (%) | 3 (2.4) | 8 (3.7) | 0.51 | 4 (3.9) | 7 (3.2) | **0.03** |
| **Leukocytes on dipsticks (%)** | 64 (50.8) | 97 (45.3) | | 35 (34.3) | 111 (50.5) | |
| **Both on dipsticks (%)** | 59 (46.8) | 107 (50.0) | | 63 (61.8) | 100 (45.5) | |

*UTI, urinary tract infection.

**Risk factors for resistance was originally measured in the clinical trial and included: systemic antibiotic exposure (at least 1 dose) or hospitalization in an acute or long term care center in the previous 12 months, UTI fulfilling criteria for healthcare-associated infection, carriage of resistant organisms in the prior 12 months, stay of at least 1 month in a high-risk country (any country in the Mediterranean basin excluding France; South or Southeast Asia; the Middle East; Africa; and Central or South America).

resistant uropathogens; seven (41%) and four (50%) of these had clinical failure and six (35%) and one (25%) microbiologic failure. Among the 438 women with available outcomes data (85%) and a positive or negative baseline culture, age>52 years and treatment with fosfomycin

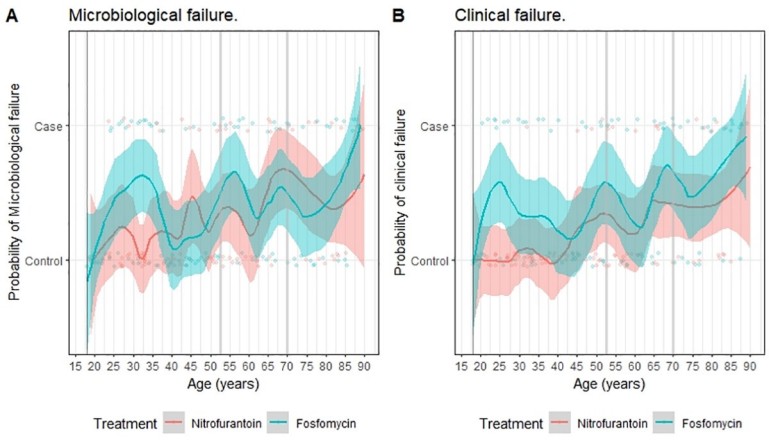

**Fig 2. Variation of clinical and microbiological failure across different age groups and treatment groups.**

**Table 2. Results of multivariate risk factor models in women in the per-protocol population with microbiologically confirmed UTI.**

| Terms | Clinical failure | | | Microbiological failure | | |
|---|---|---|---|---|---|---|
| | Odds Ratio | 95% CI | p-value | Odds Ratio | 97.5% CI | p-value |
| **Age [18;52.5]** | *Reference* | | | | | |
| **Age [52.5;105]** | 3.01 | 1.84–4.98 | **<0.001** | 2.55 | 1.54–4.25 | **<0.001** |
| **Nitrofurantoin** | *Reference* | | | | | |
| **Fosfomycin** | 2.35 | 1.47–3.8 | **<0.001** | 1.58 | 0.97–2.60 | 0.07 |
| **Centre (GE)** | *Reference* | | | | | |
| **Centre (LO)** | 1.02 | 0.59–1.76 | 0.94 | 1.58 | 0.93–2.7 | 0.09 |
| **Centre (TA)** | 3.09 | 1.61–5.98 | **<0.001** | 0.61 | 0.25–1.38 | 0.25 |

remained significantly associated with clinical failure, while a negative baseline culture was protective (OR 0.23 [95%CI 0.11–0.44], S12 Table in S1 Appendix).

Among the 328 women with available day-14 and day-28 microbiologic and clinical outcomes data, 23 had clinical success but persistent or recurrent bacteriuria by day 14. Among these, only 4 (17%) would go on to experience clinical failure by day 28 (p = .26).

## Discussion

Age > 52 years was a significant risk factor for clinical and microbiological failure in the overall cohort and in the subgroup of women with *E. coli* UTI. Although it remains significant, nitrofurantoin's clinical superiority over fosfomycin through day 28 becomes less pronounced with patients' increasing age. Other factors affecting clinical outcomes were treatment and recruitment site, the former confirming RCT findings and the latter reflecting variation among centres. Interestingly, diabetes was strongly associated with microbiological failure in univariate analysis, but not with clinical failure. Yet in adjusted analyses, the association with microbiologic failure was not retained. Among the 23 women with resolution of symptoms but persistent or recurrent bacteriuria on day 14, only 4 (17%) would proceed to clinical failure by day 28.

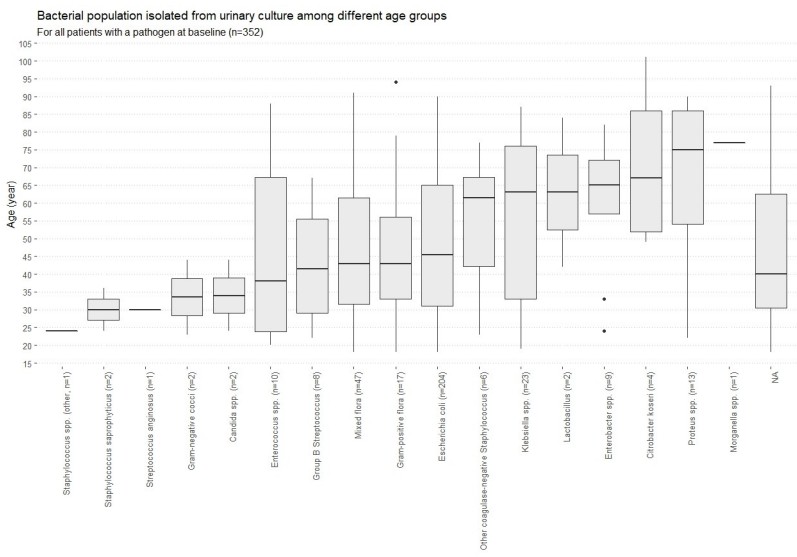

**Fig 3. Bacterial population isolated from urinary culture among different age groups.**

These findings are in line with those of Jorgensen et al., who found that age >65 years predicted a return visit to the emergency department in the 30 days following treatment for uncomplicated UTI [27], although this retrospective study relied upon coding for UTI diagnosis and included men and women. In a RCT comparing diclofenac to norfloxacin in 256 women with uncomplicated lower UTI, Kronenberg et al. [9] observed increased symptom resolution by day 3 and reduced need for later antibiotic therapy in women under 45 years, but differences were not statistically significant in this relatively small population.

The influence of age on persistence or relapse might be explained by several factors, including immunosenescence, with its gradual T-cell dysregulation and general deterioration of mucosal immunity [28], and probably by other pathways such as increased urinary stasis and post-menopausal change in the vaginal flora [29]. Considering both the risk of failure and the high incidence of symptomatic infections among elderly (12.8/100 patient years between 86–90 years) [30], this population should be considered carefully before delayed antibiotic therapy. The decline in nitrofurantoin's clinical superiority over fosfomycin with increased age requires confirmation in future trials; it is likely multifactorial and may be due to reduced renal function and other pharmacokinetic factors in the elderly, such as reduced absorption [31].

Diabetes is generally associated with more frequent UTI, primarily through immunological impairment, neurogenic bladder, and increased bladder glucose concentration [32], but in our population it was not associated with a return of symptoms, perhaps reflecting compromised local sensorimotor pathways. The association between diabetes and microbiological failure was thus expected, with association between glycemic control and asymptomatic bacteriuria already observed elsewhere [33, 34].

An increased proportion of clinical (but not bacteriologic) failure was observed among Tel Aviv participants. This centre effect could be explained by the inclusion of women with more severe illness or by cultural differences in the self-reporting of symptoms, possibly generating a reporting bias, as observed elsewhere [35].

This study has limitations, chief among them its retrospective nature: data were limited to those collected in the RCT, with no information available on concurrent renal function, additional symptomatology, sexual activity and other behavioural practices, and duration of symptoms before inclusion. Its strengths are its international and relatively large population with a sizeable event rate, improving generalization of findings to other settings, as well as its regular microbiologic follow-up data even among asymptomatic participants.

## Conclusions

Given rising antibiotic resistance and increasing use of delayed antibiotic prescription, the ability to identify early on patients at risk for failure is becoming ever more important. While they require confirmation in prospective studies, these results suggest that post-menopausal women with acute cystitis are at increased risk for treatment failure, though diabetic women are not. The low proportion of clinical failure among women with asymptomatic persistent or recurrent bacteriuria supports current recommendations to avoid treatment of asymptomatic bacteriuria (and thus surveillance cultures in general), even in those with recent symptomatic infection.

## Supporting information

**S1 Appendix.**
(DOCX)

## Acknowledgments

We thank Ms. Caroline Brossier of the Geneva University Hospitals for data collection, and Drs. Jocelyne Chabert and Khaled Mostaguir of the Center for Clinical Research, Geneva University Hospitals and Faculty of Medicine, for monitoring and data management, respectively. We also thank all trial participants and study assistants for their efforts.

## Author Contributions

**Conceptualization:** Romain Martischang, Stephan Harbarth, Angela Huttner.

**Data curation:** Romain Martischang.

**Formal analysis:** Romain Martischang.

**Investigation:** Romain Martischang, Angela Huttner.

**Methodology:** Romain Martischang, Stephan Harbarth, Angela Huttner.

**Project administration:** Romain Martischang, Stephan Harbarth, Angela Huttner.

**Resources:** Romain Martischang, Maciek Godycki-Ćwirko, Anna Kowalczyk, Katarzyna Kosiek, Adi Turjeman, Tanya Babich, Shachaf Shiber, Leonard Leibovici, Angela Huttner.

**Software:** Romain Martischang.

**Supervision:** Stephan Harbarth, Angela Huttner.

**Validation:** Stephan Harbarth, Angela Huttner.

**Visualization:** Romain Martischang, Angela Huttner.

**Writing – original draft:** Romain Martischang, Angela Huttner.

**Writing – review & editing:** Romain Martischang, Maciek Godycki-Ćwirko, Anna Kowalczyk, Katarzyna Kosiek, Adi Turjeman, Tanya Babich, Shachaf Shiber, Leonard Leibovici, Elodie von Dach, Stephan Harbarth, Angela Huttner.

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
