## [Decision Letter · Decision Letter 0]

9 Jun 2021

PONE-D-21-14320

Risk factors for treatment failure in women with uncomplicated lower urinary tract infection

PLOS ONE

Dear Dr. Martischang,

Thank you for submitting your manuscript to PLOS ONE. After careful consideration, we feel that it has merit but does not fully meet PLOS ONE’s publication criteria as it currently stands. Therefore, we invite you to submit a revised version of the manuscript that addresses the points raised during the review process.

Please revise the manuscript according to both Reviewers' comments.

We look forward to receiving your revised manuscript.

Kind regards,

Justyna Gołębiewska

Academic Editor

PLOS ONE

Journal Requirements:

Reviewers' comments:

Reviewer's Responses to Questions

**Comments to the Author**

1. Is the manuscript technically sound, and do the data support the conclusions?

Reviewer #1: Partly

Reviewer #2: Yes

2. Has the statistical analysis been performed appropriately and rigorously? 

Reviewer #1: I Don't Know

Reviewer #2: Yes

3. Have the authors made all data underlying the findings in their manuscript fully available?

Reviewer #1: Yes

Reviewer #2: Yes

4. Is the manuscript presented in an intelligible fashion and written in standard English?

Reviewer #1: Yes

Reviewer #2: Yes

5. Review Comments to the Author

Reviewer #1: The article has been well written and provides evidence for risk factors for clinical and microbiological failure of treatment in a relatively large population of women with lower UTI. However, you conclude that based on the result that postmenopausal age is a risk factor for treatment failure, this risk factor should be considered when considering delayed antibiotic therapy. I do not understand how you came to this part of the conclusion. In delayed antibiotic therapy you do not treat and you rely on spontaneous recovery within a week. In my opinion the risk in delayed therapy is that spontaneous recovery does not happen or that an upper UTI infection develops. Can you elaborate on your conclusion concerning delayed therapy?

Your study included hospitalized and ambulatory adult women with lower UTI symptoms. I expect delayed therapy to be applied only to ambulatory adult women. Do you know whether your conclusions also apply to ambulatory women only?

An ethical statement is missing.

It took me a lot of time to understand figure 1, comparing it with the numbers you mention in the Abstract and in Methods. I think it would help if you put all the main groups in the left column and the groups with clinical and microbiological failure on the right, together with the numbers of controls. In Figure 1 it seems like the 340 women with microbiologically confirmed UTI is divided into a group with clinical failure (126 and 214 controls?), a group with micromiological failure (102 with 220 controls) and women with E-coli related UTI (70 with 134 controls?) and 136 women with non-E.coli related UTI.

For the women with E.coli related UTI you only show the women with clinical failure (70 and 134 controls?) but you do not show the women with microbiological failure (71 and 128 controls?).

In the Abstract,section Methods you mention that controls were those who did not expoerience clin of microbiol failure. In lines 105 and 106 you mention that only women adhering to the RCT's protocol were included. I noticed that cases plus controls not always add up to the total number of women in a main group. In the Result section, I would expect a mention of how many cases or controls were missing because of this.

Line 195: 4 should be in line with the other numbers below 10.

Reviewer #2: This study assessed risk factors for clinical and microbiological failure in women with lower UTI via a case-control study nested within a randomized clinical trial. Postmenopausal age was an independent risk factor for clinical and microbiological treatment failure in women with lower UTI; diabetes was not associated with failure. The authors’ assessment of microbiological failure provides an understudied insight that would be of interest to a general urology and primary care audience.

Abstract and Introduction:

-Given that the authors are from Switzerland, it would be useful for them to identify (either in the abstract of Introduction) whether their cohort – and thus their recommendations – should apply to only a Switzerland demographic, or whether their research could extend outside (or not).

-Similarly, there is some evidence to suggest that women’s menopausal status may also impact of doctor’s choice of therapy and effects of that therapy. The authors do comment on this (and include age as a variable in their analyses), but a more thorough description of this would be useful in describing the demographic, as well as treatment failure (particularly as it relates to age).

Methods:

-The authors state that “only women adhering to the RCT’s protocol were included to avoid the problems of missing data/outcomes and non-adherence” – how many patients were excluded due to this? And was missing data quantified in any way? That is, did most of these patients with missing data have 1-2 missing data points vs. most of their data points missing? And did the demographics of excluded patients differ from those who were included?

-It’s unclear based on the author’s current description why a cutoff of 52 years was chosen and this could use further explanation, particularly given that it is critical to the main results and interpretation.

-What statistical analysis software was used?

Results:

-Although age was entered based on the 52.5 cutoff, it might be worth classifying women as pre-menopausal or post-menopausal and including that as a variable in the models, if that data was collected by the authors.

-Table 2 is quite helpful, but it would be useful to see the full models (all variables entered).

Discussion:

-The authors note that recruitment site (and variation among centres) impact clinical outcomes. Why was this the case? Additional explanation is needed for results to be interpretable.

-Given that the finding that diabetes was strongly associated with microbiological failure, more discussion should be included. Was this predicted/hypothesized? If yes, why or why not?

-When discussing limitations, the authors note that strength of their study was that it was multi-centre; however, these centres clearly varied (given recruitment site impacted outcomes). Why might that be more of a limitation? – the authors might want to discuss this a bit more.

6. PLOS authors have the option to publish the peer review history of their article (what does this mean?). If published, this will include your full peer review and any attached files.

Reviewer #1: No

Reviewer #2: No

---

## [Author Response · Author response to Decision Letter 0]

29 Jun 2021

On behalf of the authors, we would like to thank the reviewers for their careful reading and valuable, in-depth comments. We provide point-by-point responses to specific reviewers and editor in the document attached and labelled "Response to Reviewers". 

Furthermore, we have ensured that our manuscript meets PLOS ONE’s style requirements, and have made the minimal anonymized data set available to the best of our ability on Zenodo (https://zenodo.org/, DOI: 10.5281/zenodo.5024262). We used data collected from Geneva (Switzerland), Lodz (Poland), and Tel Aviv (Israel). By Swiss and Polish laws, there is no restriction to share anonymized information. However, the situation is more complicated for Israeli data. Additional approvals from both the Ethics Committee and the Data Privacy Committee would be required, and in the experience of our co-investigator, Prof. Leonard Leibovici, are unlikely to be granted, as his organization does not typically allow ‘placing its patients’ data into public view’, even when anonymized. We will thus be able to provide only the Swiss and Polish dataset; we hope that you will deem this sufficient. If individual researchers want to recycle the complete dataset, we will offer them to contact us any time to find an appropriate solution. 

After inclusion of multiple details in the revised manuscript as requested by the reviewers, the MS length has increased to 2’172 words (text only), despite all attempts to shorten the main text.

Kind regards, 

Romain Martischang

---

## [Decision Letter · Decision Letter 1]

9 Aug 2021

Risk factors for treatment failure in women with uncomplicated lower urinary tract infection

PONE-D-21-14320R1

Dear Dr. Martischang,

We’re pleased to inform you that your manuscript has been judged scientifically suitable for publication and will be formally accepted for publication once it meets all outstanding technical requirements.

Kind regards,

Justyna Gołębiewska

Academic Editor

PLOS ONE

Additional Editor Comments (optional):

Reviewers' comments:

Reviewer's Responses to Questions

**Comments to the Author**

1. If the authors have adequately addressed your comments raised in a previous round of review and you feel that this manuscript is now acceptable for publication, you may indicate that here to bypass the “Comments to the Author” section, enter your conflict of interest statement in the “Confidential to Editor” section, and submit your "Accept" recommendation.

Reviewer #1: All comments have been addressed

Reviewer #2: All comments have been addressed

2. Is the manuscript technically sound, and do the data support the conclusions?

Reviewer #1: (No Response)

Reviewer #2: Yes

3. Has the statistical analysis been performed appropriately and rigorously? 

Reviewer #1: (No Response)

Reviewer #2: Yes

4. Have the authors made all data underlying the findings in their manuscript fully available?

Reviewer #1: (No Response)

Reviewer #2: Yes

5. Is the manuscript presented in an intelligible fashion and written in standard English?

Reviewer #1: (No Response)

Reviewer #2: Yes

6. Review Comments to the Author

Reviewer #1: (No Response)

Reviewer #2: The authors did a great job address reviewer comments and feedback. With these revisions, I recommend the manuscript be accepted for publication.

7. PLOS authors have the option to publish the peer review history of their article (what does this mean?). If published, this will include your full peer review and any attached files.

Reviewer #1: No

Reviewer #2: No

---

## [Editor Report · Acceptance letter]

13 Aug 2021

PONE-D-21-14320R1 

Risk factors for treatment failure in women with uncomplicated lower urinary tract infection 

Dear Dr. Martischang:

I'm pleased to inform you that your manuscript has been deemed suitable for publication in PLOS ONE. Congratulations! Your manuscript is now with our production department. 

Kind regards, 

on behalf of

Dr. Justyna Gołębiewska 

Academic Editor

PLOS ONE